# Traversing the Local Polytopes of ReLU Neural Networks

**Shaojie Xu, Joel Vaughan, Jie Chen, Aijun Zhang, Agus Sudjianto**

Wells Fargo & Company *

## Abstract

Although neural networks (NNs) with ReLU activation functions have found success in a wide range of applications, their adoption in risk-sensitive settings has been limited by the concerns on robustness and interpretability. Previous works to examine robustness and to improve interpretability partially exploited the piecewise linear function form of ReLU NNs. In this paper, we explore the unique topological structure that ReLU NNs create in the input space, identifying the adjacency among the partitioned local polytopes and developing a traversing algorithm based on this adjacency. Our polytope traversing algorithm can be adapted to a wide range of applications related to robustness and interpretability. As the traversing algorithm explicitly visits all local polytopes, it returns a clear and full picture of the network behavior within the traversed region. The time and space complexity of the traversing algorithm is determined by the number of a ReLU NN's partitioning hyperplanes passing through the traversing region.

## 1   Introduction & Related Work

Neural networks with rectified linear unit activation functions (ReLU NNs) are arguably the most popular type of neural networks in deep learning. This type of network enjoys many appealing properties including better performance than NNs with sigmoid activation (Glorot, Bordes, and Bengio 2011), universal approximation ability (Arora et al. 2018; Lu et al. 2017; Montufar et al. 2014; Schmidt-Hieber 2020), and fast training speed via scalable algorithms such as stochastic gradient descent (SGD) and its variants (Zou et al. 2020).

Despite their strong predictive power, ReLU NNs have seen limited adoption in risk-sensitive settings (Bunel et al. 2018). These settings require the model to make robust predictions against potential adversarial noise in the input (Athalye et al. 2018; Carlini and Wagner 2017; Goodfellow, Shlens, and Szegedy 2014; Szegedy et al. 2014). The alignment between model behavior and human intuition is also desirable (Liu et al. 2019): prior knowledge such as monotonicity may be incorporated into model design and training (Daniels and Velikova 2010; Gupta et al. 2019; Liu et al. 2020; Sharma and Wehrheim 2020); users and auditors of

the model may require a certain degree of explanations of the model predictions (Gopinath et al. 2019; Chu et al. 2018).

The requirements in risk-sensitive settings has motivated a great amount of research on verifying certain properties of ReLU NNs. These works often exploit the piecewise linear function form of ReLU NNs. In Bastani et al. (2016) the robustness of a network is verified in very small input region via linear programming (LP). To consider the nonlinearity of ReLU activation functions, Ehlers (2017); Katz et al. (2017); Pulina and Tacchella (2010, 2012) formulated the robustness verification problem as a satisfiability modulo theories (SMT) problem. A more popular way to model ReLU nonlinearity is to introduce a binary variable representing the on-off patterns of ReLU neurons. Property verification can then be solved using mixed-integer programming (MIP) (Anderson et al. 2020; Fischetti and Jo 2017; Liu et al. 2020; Tjeng, Xiao, and Tedrake 2018; Weng et al. 2018).

The piecewise linear functional form of ReLU NNs also creates distinct topological structures in the input space. Previous studies have shown that a ReLU NN partitions the input space into convex polytopes and has one linear model associated with each polytope (Montufar et al. 2014; Serra, Tjandraatmadja, and Ramalingam 2018; Croce, Andriushchenko, and Hein 2019; Robinson, Rasheed, and San 2019; Sudjianto et al. 2020; Yang et al. 2020). Each polytope can be coded by a binary activation code, which reflects the on-off patterns of the ReLU neurons. The number of local polytopes is often used as a measure of the model's expressivity (Hanin and Rolnick 2019; Lu et al. 2017). Built upon this framework, multiple studies (Sudjianto et al. 2020; Yang, Zhang, and Sudjianto 2020; Zhao et al. 2021) tried to explain the behavior of ReLU NNs and to improve their interpretability. They viewed ReLU NN as a collection of linear models. However, the relationship among the local polytopes and their linear models was not fully investigated.

When the network's behavior within some specific region in the input space is of interest, one can collect all the local polytopes overlapped with the region to conduct analysis. The methods to collect these polytopes can be categorized into top-down and bottom-up approaches. The top-down approaches in Xiang, Tran, and Johnson (2017); Yang et al. (2020) pass the entire region of interest through a ReLU NN and calculate how the hyperplanes corresponding to the neurons partition the region into local polytopes. The major

---

*The views expressed in the paper are those of the authors and do not represent the views of Wells Fargo.

drawback of the top-down approach is that the analysis must start after the computationally expensive forward passing is fully finished.

One the contrary, the bottom-up approaches start from a point of interest inside the region, moving from one local polytope to another while running the analysis, and can be stopped at any time. Croce and Hein (2018); Croce, Rauber, and Hein (2020) achieved the movement among polytopes by generating a sequence of samples in the input space using randomized local search. Although being computationally simple, this sample-based method does not guarantee covering all polytopes inside the region of interest. The most recent work and also the closest to ours is Vincent and Schwager (2021), where polytope boundaries and adjacency are identified using LP, and the traversing is done directly on the polytopes.

In this paper, we explore the topological relationship among the local polytopes created by ReLU NNs. We propose algorithms to identify the adjacency among these polytopes, based on which we develop traversing algorithms to visit all polytopes within a bounded region in the input space. Compared with (Vincent and Schwager 2021), our polytope traversing algorithm exploits ReLU NNs' hierarchical partitioning of the input space to reduce computational overhead and achieve significant acceleration for discovering adjacent polytopes. The thoroughness of our traversing algorithm is proved. The algorithm is versatile and can be adapted to many applications related to model robustness and interpretability. Although we focus on ReLU NN with fully connected layers through out this paper, our polytope traversing algorithm can be naturally extended to other piecewise linear networks such as those containing convolutional and maxpooling layers.

The rest of this paper is organized as follows: Section 2 reviews how polytopes are created by ReLU NNs. Section 3 introduces two related concepts: the boundaries of a polytope and the adjacency among the polytopes. Our polytope traversing algorithm is described in Section 4. Section 5 discusses several applications that our traversing algorithm can be adapted to. The paper is concluded in Section 6.

## 2 The Local Polytopes in ReLU NNs

### 2.1 The case of one hidden layer

A ReLU NN partitions the input space $\mathbb{R}^P$ into several polytopes and forms a linear model within each polytope. To see this, we first consider a simple NN with one hidden layer of $M$ neurons. It takes an input $\boldsymbol{x} \in \mathbb{R}^P$ and outputs $\boldsymbol{o} \in \mathbb{R}^Q$ by calculating:

$$\boldsymbol{o} = \boldsymbol{W}^o \boldsymbol{h} + \boldsymbol{b}^o = \boldsymbol{W}^o \left( \sigma(\boldsymbol{W}\boldsymbol{x} + \boldsymbol{b}) \right) + \boldsymbol{b}^o$$

$$\text{where } \sigma(\boldsymbol{x})_m = \begin{cases} 0, & \boldsymbol{x}_m < 0 \\ \boldsymbol{x}_m, & \boldsymbol{x}_m \geq 0 \end{cases} . \quad (1)$$

For problems with a binary or categorical target variable (i.e. binary or multi-class classification), a sigmoid or softmax layer is added after $\boldsymbol{o}$ respectively to convert the convert the NN outputs to proper probabilistic predictions.

The ReLU activation function $\sigma(\cdot)$ inserts non-linearity into the model by checking a set of linear inequalities: $\boldsymbol{w}_m^T \boldsymbol{x} + b_m \geq$

$0, \ m = 1, 2, \ldots, M$, where $\boldsymbol{w}_m^T$ is the $m$th row of matrix $\boldsymbol{W}$ and $b_m$ is the $m$th element of $\boldsymbol{b}$. Each neuron in the hidden layer creates a **partitioning hyperplane** in the input space with the linear equation $\boldsymbol{w}_m^T \boldsymbol{x} + b_m = 0$. The areas on two sides of the hyperplane are two **halfspaces**. The entire input space is, therefore, partitioned by these $M$ hyperplanes. We define a **local polytope** as a set containing all points that fall on the same side of each and every hyperplane. The polytope encoding function (2) uses an element-wise indicator function $\mathbb{1}(\cdot)$ to create a unique binary code $\boldsymbol{c}$ for each polytope. Since the $m$th neuron is called "ON" for some $\boldsymbol{x}$ if $\boldsymbol{w}_m^T \boldsymbol{x} + \tilde{b}_m \geq 0$, the code $\boldsymbol{c}$ also represents the on-off pattern of the neurons. Using the results of this encoding function, we can express each polytope as an intersection of $M$ halfspaces as in (3), where the binary code $\boldsymbol{c}$ controls the directions of the inequalities.

$$C(\boldsymbol{x}) = \mathbb{1}(\boldsymbol{W}\boldsymbol{x} + \boldsymbol{b} \geq 0) . \quad (2)$$

$$\mathcal{R}_{\boldsymbol{c}} = \{ \boldsymbol{x} \mid (-1)^{c_m}(\boldsymbol{w}_m^T \boldsymbol{x} + b_m \leq 0), \ \forall m = 1, \ldots, M \} . \quad (3)$$

Figure 1.(b) shows an example of ReLU NN trained on a two-dimensional synthetic dataset (plotted in Figure 1.(a)). The bounded input space is $[-1, 1]^2$ and the target variable is binary. The network has one hidden layer of 20 neurons. The partitioning hyperplanes associated with these neurons are plotted as the blue dashed lines. They form in total 91 local polytopes within the bounded input space.

For a given $\boldsymbol{x}$, if $\boldsymbol{w}_m^T \boldsymbol{x} + b_m \geq 0$, the ReLU neuron turns on and passes through the value. Otherwise, the neuron is off and suppresses the value to zero. Therefore, if we know the $m$th neuron is off, we can mask the corresponding $\boldsymbol{w}_m$ and $b_m$ by zeros and create $\tilde{\boldsymbol{W}}_{\boldsymbol{c}}$ and $\tilde{\boldsymbol{b}}_{\boldsymbol{c}}$ that satisfy (5). The non-linear operation, therefore, can be replaced by the a locally linear operation after zero-masking. Because each local polytope $\mathcal{R}_{\boldsymbol{c}}$ has a unique neuron activation pattern encoded by $\boldsymbol{c}$, the zero-masking process in (4) is also unique for each polytope. Here, $\boldsymbol{1}$ is a vector of 1s of length $p$ and $\otimes$ denotes element-wise product.

$$\tilde{\boldsymbol{W}}_{\boldsymbol{c}} = \boldsymbol{W} \otimes (\boldsymbol{c}\boldsymbol{1}^T) , \ \tilde{\boldsymbol{b}}_{\boldsymbol{c}} = \boldsymbol{b} \otimes \boldsymbol{c} , \quad (4)$$

$$\sigma(\boldsymbol{W}\boldsymbol{x} + \boldsymbol{b}) = \tilde{\boldsymbol{W}}_{\boldsymbol{c}}\boldsymbol{x} + \tilde{\boldsymbol{b}}_{\boldsymbol{c}}, \quad \forall \boldsymbol{x} \in \mathcal{R}_{\boldsymbol{c}} . \quad (5)$$

Within each polytope, as the non-linearity is taken out by the zero-masking process, the input $\boldsymbol{x}$ and output $\boldsymbol{o}$ have a linear relationship:

$$\boldsymbol{o} = \boldsymbol{W}^o(\sigma(\boldsymbol{W}\boldsymbol{x} + \boldsymbol{b})) + \boldsymbol{b}^o = \hat{\boldsymbol{W}}_{\boldsymbol{c}}^o \boldsymbol{x} + \hat{\boldsymbol{b}}_{\boldsymbol{c}}^o , \ \forall \boldsymbol{x} \in \mathcal{R}_{\boldsymbol{c}} ,$$

$$\text{where } \hat{\boldsymbol{W}}_{\boldsymbol{c}}^o = \boldsymbol{W}^o \tilde{\boldsymbol{W}}_{\boldsymbol{c}} , \ \hat{\boldsymbol{b}}_{\boldsymbol{c}}^o = \boldsymbol{W}^o \tilde{\boldsymbol{b}}_{\boldsymbol{c}} + \boldsymbol{b}^o \quad (6)$$

The linear model associated with polytope $\mathcal{R}_{\boldsymbol{c}}$ has the weight matrix $\hat{\boldsymbol{W}}_{\boldsymbol{c}}^o$ and the bias vector $\hat{\boldsymbol{b}}_{\boldsymbol{c}}^o$. The ReLU NN is now represented by a collection of linear models, each defined on a local polytope $\mathcal{R}_{\boldsymbol{c}}$.

In Figure 1.(b), we represent the linear model in each local polytopes as a red solid line indicating $(\hat{\boldsymbol{w}}_{\boldsymbol{c}}^o)^T \boldsymbol{x} + \hat{b}_{\boldsymbol{c}}^o = 0$. In this binary response case, the two sides of this line have the opposite class prediction. We only plot the line if it passes through its corresponding polytope. For other polytopes, the entire polytopes fall on one side of their corresponding class-separating lines and the predicted class is the same within the whole polytope. The red lines all together form the decision boundary of the ReLU NN and are continuous when passing through one polytope to another. This is a direct result of ReLU NN being a continuous model.

### 2.2 The case of multiple layers

We can generalize the results to ReLU NNs with multiple hidden layers. A ReLU NN with $L$ hidden layers hierarchically partitions the input space and is locally linear in each and every **level-$L$ polytope**. Each level-$L$ polytope $\mathcal{R}^L$ has a unique binary code

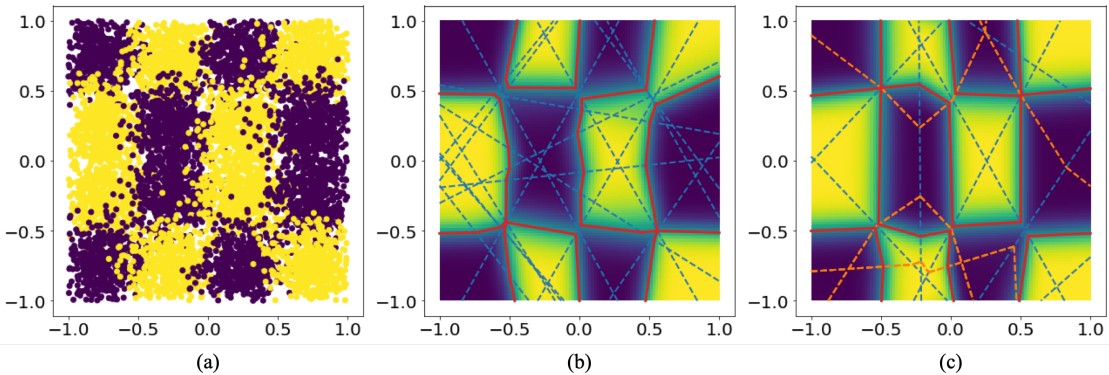

Figure 1: Examples of trained ReLU NNs and their local polytopes. (a) The grid-like training data with binary target variable. (b) A trained ReLU NN with one hidden layer of 20 neurons. The heatmap shows the predicted probability of a sample belong to class 1. The blue dashed lines are the partitioning hyperplanes associated with the ReLU neurons, which form 91 local polytopes in total. The red solid lines represent the linear model within each polytope where class separation occurs. (c) A trained ReLU NN with two hidden layers of 10 and 5 neurons respectively. The blue dashed lines are the partitioning hyperplanes associated with the first 10 ReLU neurons, forming 20 level-1 polytopes. The orange dashes lines are the partitioning hyperplanes associated with the second 5 ReLU neurons within each level-1 polytope. There are in total 41 (level-2) local polytopes. The red solid lines represent the linear model within each level-2 polytope where class separation occurs.

$c^1 c^2 \ldots c^L$ representing the activation pattern of the neurons in all $L$ hidden layers. The corresponding partitioning hyperplanes of each level, $\hat{W}^l x + \hat{b}^l = 0$, $l = 1, 2, \ldots, L$, can be calculated recursively level by level, using the zero masking procedure:

$$\hat{W}^1 = W^1 \,,\ \hat{b}^1 = b^1 \tag{7}$$

$$\tilde{W}^l = \hat{W}^l \otimes (c^l \mathbf{1}^T) \,,\ \tilde{b}^l = \hat{b}^l \otimes c^l \tag{8}$$

$$\hat{W}^{l+1} = W^{l+1} \tilde{W}^l \,,\ \hat{b}^{l+1} = W^{l+1} \tilde{b}^l + b^{l+1} \,. \tag{9}$$

We emphasis that $\tilde{W}^l, \tilde{b}^l, \hat{W}^{l+1}$, and $\hat{b}^{l+1}$ depend on all polytope code up to level $l$: $c^1 c^2 \ldots c^l$. The subscription $c$ is dropped to simplify the notations.

At each level $l$, the encoding function $C^l(\cdot)$ and the polytope $\mathcal{R}^l$ expressed as an intersection of $\sum_{t=1}^l M_t$ halfspaces can be written recursively as:

$$C^1(x) = \mathbb{1}(W^1 x + b^1 \geq 0) \tag{10}$$

$$\mathcal{R}^1 = \{x \mid (-1)^{c^m} \left( (w^1)_m^T x + (b^1)_m \leq 0 \right), \\ \forall m = 1, 2, \ldots, M_1\} \tag{11}$$

$$C^{l+1}(x) = \mathbb{1}(\hat{W}^{l+1} x + \hat{b}^{l+1} \geq 0) \,,\ \forall x \in \mathcal{R}^l \tag{12}$$

$$\mathcal{R}^{l+1} = \{x \mid (-1)^{c^m} \left( (\hat{w}^{l+1})_m^T x + (\hat{b}^{l+1})_m \leq 0 \right), \\ \forall m = 1, 2, \ldots, M_{l+1}\} \cap \mathcal{R}^l \,. \tag{13}$$

Finally, the linear model in a level-$L$ polytope is:

$$o = \hat{W}^o x + \hat{b}^o \,,\ \forall x \in \mathcal{R}^L \,, \\ \text{where } \hat{W}^o = W^o \tilde{W}^L \,,\ \hat{b}^o = W^o \tilde{b}^L + b^o \,. \tag{14}$$

Figure 1.(c) shows an example of ReLU NN with two hidden layers of size 10 and 5 respectively. The partitioning hyperplanes associated with the first 10 neuron are plotted as the blue dashed lines. They form 20 level-1 polytopes within the bounded input space. Within each of the level-1 polytope, the hyperplanes associated with the second 5 neurons further partition the polytope. In many cases, some of the 5 hyperplanes are outside the level-1

polytope and, therefore, not creating a new sub-partition. The hyperplanes do create new partitions are plotted as the orange dashed lines. The orange lines are only straight within a level-1 polytope but are continuous when passing through one polytope to another, which is also a result of ReLU NN being a continuous model. In total, this ReLU NN creates 41 (level-2) local polytopes. As in Figure 1.(b), the linear model within each level-2 polytope is represented as a red solid line if class separation occurs within the polytope.

## 3 Polytope Boundaries and Adjacency

Beyond viewing ReLU NNs as a collection of linear models defined on local polytopes, we explore the topological relationship among these polytopes. A key concept is the **boundaries** of each polytope. As shown in (13), each level-$l$ polytope $\mathcal{R}_c$ with corresponding binary code $c = c^1 c^2 \ldots c^l$ is an intersection of $\sum_{t=1}^l M_t$ halfspaces induced by a set of inequality constraints. Two situations can rise among these inequalities. First, an arbitrary $c$ may lead to conflicting inequalities and makes $\mathcal{R}_c$ an empty set. This situation can be common when the number of neurons is much larger than the dimension of the input space. Second, there can be **redundant inequalities** which means removing them does not affect set $\mathcal{R}_c$. We now show that the non-redundant inequalities are closely related to the boundaries of a polytope.

**Definition 3.1** *Let $\mathcal{R}$ contains all $x \in \mathbb{R}^P$ that satisfy $M$ linear inequalities: $\mathcal{R} = \{x | g_1(x) \leq 0, g_2(x) \leq 0, \ldots, g_M(x) \leq 0\}$. Assume that $\mathcal{R} \neq \emptyset$. Let $\tilde{\mathcal{R}}$ contains all $x$'s that satisfy $M-1$ linear inequalities: $\tilde{\mathcal{R}} = \{x | g_1(x) \leq 0, \ldots, g_{m-1}(x) \leq 0, g_{m+1}(x) \leq 0, \ldots, g_M(x) \leq 0\}$. Then the inequality $g_m(x) \leq 0$ is a **redundant inequality** with respect to (w.r.t.) $\mathcal{R}$ if $\mathcal{R} = \tilde{\mathcal{R}}$.*

With the redundant inequality defined above, the following lemma provides an algorithm to identify them. The proof of this lemma is in the Appendix.

**Lemma 3.1** *Given a set $\mathcal{R} = \{x | g_1(x) \leq 0, \ldots, g_M(x) \leq 0\} \neq \emptyset$, then $g_m(x)$ is a redundant inequality if the new set formed by flipping this inequality is empty: $\hat{\mathcal{R}} = \{x | g_1(x) \leq 0, \ldots, g_m(x) \geq 0, \ldots, g_M(x) \leq 0\} = \emptyset$.*

We can now define the boundaries of a polytope formed by a set of linear inequalities using a similar procedure in Lemma 3.1. The

concept of polytope boundaries also leads to the definition of adjacency. Intuitively, we can move from one polytope to its adjacent polytope by crossing a boundary.

**Definition 3.2** *Given a non-empty set formed by $M$ linear inequalities: $\mathcal{R} = \{\boldsymbol{x}|g_1(\boldsymbol{x}) \leq 0, \ldots, g_M(\boldsymbol{x}) \leq 0\} \neq \emptyset$, then the hyperplane $g_m(\boldsymbol{x}) = 0$ is a **boundary** of $\mathcal{R}$ if the new set formed by flipping the corresponding inequality is non-empty: $\hat{\mathcal{R}} = \{\boldsymbol{x}|g_1(\boldsymbol{x}) \leq 0, \ldots, g_m(\boldsymbol{x}) \geq 0, \ldots, g_M(\boldsymbol{x}) \leq 0\} \neq \emptyset$. Polytope $\hat{\mathcal{R}}$ is called **one-adjacent** to $\mathcal{R}$.*

Since for each polytope the directions of its linear inequalities are reflected by the binary code, two one-adjacent polytopes must have their code differ by one bit. Figure 2.(a) demonstrates the adjacency among the local polytopes. The ReLU NN is the same as in Figure 1.(b). Using the procedure in Definition 3.2, 4 out of the 20 partitioning hyperplanes are identified as the boundaries of polytope No.0 and marked in red. The 4 one-adjacent neighbors to polytope No.0 are No.1, 2, 3, and 4; each can be reached by crossing one boundary.

As we have shown in the Section 2.2, ReLU NNs create polytopes level by level. We follow the same hierarchy to define the polytope adjacency. Assume two non-empty level-$l$ polytopes, $\mathcal{R}$ and $\hat{\mathcal{R}}$, are inside the same level-$(l-1)$ polytope, which means their corresponding code $\boldsymbol{c} = \boldsymbol{c}^1\boldsymbol{c}^2\ldots\boldsymbol{c}^l$ and $\hat{\boldsymbol{c}} = \boldsymbol{c}^1\boldsymbol{c}^2\ldots\hat{\boldsymbol{c}}^l$ only differs at level-$l$. We say that polytope $\hat{\mathcal{R}}$ is a **level-$l$ one-adjacent neighbor** of $\mathcal{R}$ if $\hat{\boldsymbol{c}}^l$ and $\boldsymbol{c}^l$ only differs in one bit.

The condition that $\boldsymbol{c} = \boldsymbol{c}^1\boldsymbol{c}^2\ldots\boldsymbol{c}^l$ and $\hat{\boldsymbol{c}} = \boldsymbol{c}^1\boldsymbol{c}^2\ldots\hat{\boldsymbol{c}}^l$ only differs at level-$l$ is important. In this way, the two linear inequalities associated with each pair of bits in $\boldsymbol{c}$ and $\hat{\boldsymbol{c}}$ have the same coefficients, and the difference in $\boldsymbol{c}^l$ and $\hat{\boldsymbol{c}}^l$ only changes the direction of the linear inequality. On the other hand, if the two codes differ at a level $l' < l$, then according to the recursive calculation in (8) and (9), the codes starting from level $l'+1$ will correspond to linear inequalities of different coefficients, leaving our Definition 3.2 of adjacency not applicable.

Figure 2.(b) demonstrates the hierarchical adjacency among the local polytopes. The ReLU NN is the same as in Figure 1.(c). Level-1 polytopes $(1, \cdot)$ and $(2, \cdot)$ are both (level-1) one-adjacent to $(0, \cdot)$. Within the level-1 polytope $(0, \cdot)$, Level-2 polytopes $(0, 0)$ and $(0, 1)$ are (level-2) one-adjacent to each other. Similarly, we can identify the level-2 adjacency of the other two pairs $(1, 0) - (1, 1)$ and $(2, 0) - (2, 1)$. Note that in the plot, even thought one can move from polytope $(2, 1)$ to $(0, 1)$ by crossing one partitioning hyperplane, we do not define these two polytopes as adjacent, as they lie into two different level-1 polytopes.

# 4 Polytope Traversing

## 4.1 The case of one hidden layer

The adjacency defined in the previous section provides us an order to traverse the local polytopes: starting from an initial polytope $\mathcal{R}$, visiting its all one-adjacent neighbors, then visiting all the neighbors' neighbors and so on.

This algorithm can be viewed as breath-first search (BFS) on a **polytope graph**. To create this graph, we turn each polytope created by the ReLU NN into a node. An edge is added between each pair of polytopes that are one-adjacent to each other. The BFS algorithm uses a queue to keep track the traversing progress. At the beginning of traversing, the initial polytope is added to an empty queue and is marked as visited afterwards. In each iteration, we pop out the first polytope from the queue and identify all of its one-adjacent neighbors. Among these identified polytopes, we add those that have not been visited to the back of the queue and mark them as visited. The iteration stops when the queue is empty.

The key component of the polytope traversing algorithm is to identify a polytope's one-adjacent neighbors. For a polytope $\mathcal{R}_{\boldsymbol{c}}$ coded by $\boldsymbol{c}$ of $M$ bits, there are at most $M$ one-adjacent neighbors with codes corresponding to flipping one of the bits in $\boldsymbol{c}$. Each valid one-adjacent neighbor must be non-empty and can be reached by crossing a boundary. Therefore, we can check each linear inequality in (3) and determine whether it is a boundary or redundant. Some techniques of identifying redundant inequalities are summarized in Telgen (1983). By flipping the bits corresponding to the identified boundaries, we obtain the codes of the one-adjacent polytopes.

Equivalently, we can identify the one-adjacent neighbors by going through all $M$ candidate codes and selecting those corresponding to non-empty sets. Checking the feasibility of a set constrained by a set of linear inequalities is often referred to as the "Phase-I Problem" of LP and can be solved efficiently by modern LP solvers. During BFS iterations, we can hash the checked codes to avoid checking them repetitively. The BFS-based polytope traversing algorithm is summarized in Algorithm 1. We now state the correctness of this algorithm with its proof in Appendix.

**Theorem 4.1** *Given a ReLU NN with one hidden layer of $M$ neurons as specified in (1), Algorithm 1 covers all non-empty local polytopes created by the neural network. That is, for all $\boldsymbol{x} \in \mathbb{R}^P$, there exists one $\mathcal{R}_{\boldsymbol{c}}$ as defined in (3) such that $\boldsymbol{x} \in \mathcal{R}_{\boldsymbol{c}}$ and $\boldsymbol{c} \in \mathcal{S}_R$, where $\mathcal{S}_R$ is the result returned by Algorithm 1.*

Algorithm 1 visits all the local polytopes created by a ReLU NN within $\mathbb{R}^P$. The time complexity is exponential to the number of neurons, as all $2^M$ possible activation patterns are checked once in the worst-case scenario. The space complexity is also exponential to the number of neurons as we hash all the checked activation patterns. Furthermore, for each activation pattern, we solve a phase-I problem of LP with $M$ inequalities in $\mathbb{R}^P$. Traversing all local polytopes in $\mathbb{R}^P$, therefore, becomes intractable for neural networks with a large number of neurons.

Fortunately, traversing in $\mathbb{R}^P$ is usually undesirable. Firstly, a neural network may run into extrapolation issues for points outside the sample distribution. The polytopes far away from the areas covered by the samples are often considered unreliable. Secondly, many real-life applications, to be discussed in Section 5, only require traversing within small bounded regions to examine the local behavior of a model. In the next section, we introduce a technique to improve the efficiency when traversing within a bounded region.

## 4.2 Polytope traversing within a bounded region

We first consider a region with each dimension bounded independently: $l_j \leq x_j \leq u_j, j = 1, 2, \ldots, P$. These $2 \times P$ linear inequalities creates a hypercube denoted as $\mathcal{B}$. During the BFS-based polytope traversing, we repetitively flip the direction of one of the $M$ inequalities to identify the one-adjacent neighbors. When the bounded region is small, it is likely that only a small number of the $M$ hyperplanes cut through the hypercube. For the other hyperplanes, the entire hypercube falls onto only one side. Flipping to the other sides of these hyperplanes would leave the bounded region. Therefore, at the very beginning of polytope traversing, we can run through the $M$ hyperplanes to identify those cutting through the hypercube. Then in each neighbor identifying step, we only flip these hyperplanes.

To identify the hyperplanes cutting through the hypercube, we denote the two sides of a hyperplane $\mathcal{H}$ and $\bar{\mathcal{H}}$: $\mathcal{H} = \{\boldsymbol{x}|\boldsymbol{w}_m^T\boldsymbol{x} + b_m \leq 0\}$ and $\bar{\mathcal{H}} = \{\boldsymbol{x}|\boldsymbol{w}_m^T\boldsymbol{x} + b_m \geq 0\}$. If neither $\mathcal{H} \cap \mathcal{B}$ nor $\hat{\mathcal{H}} \cap \mathcal{B}$ is empty, we say the hyperplane $\boldsymbol{w}_m^T\boldsymbol{x} + b_m = 0$ cuts through $\mathcal{B}$. $\mathcal{H} \cap \mathcal{B}$ and $\hat{\mathcal{H}} \cap \mathcal{B}$ are both constrained by $2 \times P + 1$ inequalities, checking their feasibility can again be formulated as a

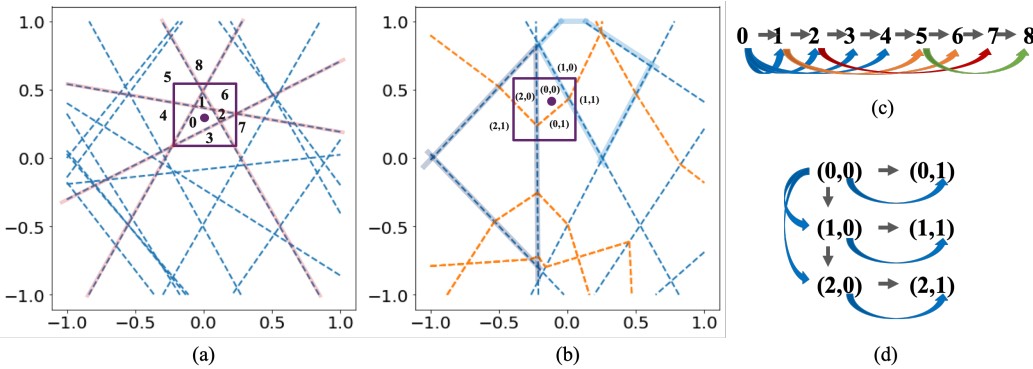

Figure 2: Demonstration of the BFS-base polytope traversing algorithm. (a) Traversing the 8 local polytopes within the bounded regions. The ReLU NN is the same as in Figure 1.(b). The lines marked in red are the boundaries of polytope No.0. (b) Traversing the 6 local polytopes within the bounded region. The ReLU NN is the same as in Figure 1.(c). The polytopes are indexed as "(level-1, level-2)". (c) The evolution of the BFS queue for traversing the local polytopes in (a). The gray arrows show the traversing order. The colored arrows at the bottom indicate the one-adjacent neighbors added to the queue. (d) The evolution of the hierarchical BFS queue for traversing the local polytopes in (b). The level-1 BFS queue is shown vertically while the level-2 BFS queue is shown horizontally.

---

**Algorithm 1: BFS-Based Polytope Traversing**

---

**Require:** A ReLU NN with one hidden layer of $M$ neurons as specified in (1).
**Require:** An initial point $\boldsymbol{x} \in \mathbb{R}^P$.
1: Initialize an empty queue $\mathcal{Q}$ for BFS.
2: Initialize an empty set $\mathcal{S}_R$ to store the codes of all visited polytopes.
3: Initialize an empty set $\mathcal{S}_c$ to store all checked codes.
4: Calculate $\boldsymbol{x}$'s initial polytope code $\boldsymbol{c}$ using (2).
5: Append $\boldsymbol{c}$ to the end of the $\mathcal{Q}$.
6: Add $\boldsymbol{c}$ to both $\mathcal{S}_R$ and $\mathcal{S}_c$.
7: **while** $\mathcal{Q}$ is not empty **do**
8:     Pop out the first element in the front of BFS queue: $\boldsymbol{c} = \mathcal{Q}.\text{pop}()$.
9:     **for** $m = 1, 2, \ldots, M$ **do**
10:         Create a candidate polytope code $\hat{\boldsymbol{c}}$ by flipping one bit in $\boldsymbol{c}$: $\hat{c}_m = 1 - c_m$ and $\hat{c}_k = c_k \forall k \neq m$.
11:         **if** $\hat{\boldsymbol{c}} \notin \mathcal{S}_c$ **then**
12:             Check if $\mathcal{R}_{\hat{\boldsymbol{c}}} = \{\boldsymbol{x}|(-1)^{\hat{c}_k}\left(\boldsymbol{w}_k^T \boldsymbol{x} + b_k\right) \leq 0,\ k = 1, 2 \ldots, M\}$ is empty using LP.
13:             Add $\hat{\boldsymbol{c}}$ to $\mathcal{S}_c$.
14:             **if** $\mathcal{R}_{\hat{\boldsymbol{c}}} \neq \emptyset$ **then**
15:                 Append $\hat{\boldsymbol{c}}$ to the end of the $\mathcal{Q}$.
16:                 Add $\hat{\boldsymbol{c}}$ to $\mathcal{S}_R$.
17: Return $\mathcal{S}_R$.

---

**Algorithm 2: Hyperplane Pre-Screening**

---

**Require:** A set of hyperplanes $\boldsymbol{w}_m^T \boldsymbol{x} + b_m \leq 0$, $m = 1, 2, \ldots, M$.
**Require:** A bounded traversing region $\mathcal{B}$, e.g. $\{\boldsymbol{x}|l_j \leq x_j \leq u_j, j = 1, 2, \ldots, P\}$.
1: Initialize an empty set $\mathcal{T}$ to store all hyperplanes cutting through $\mathcal{B}$.
2: **for** $m = 1, 2, \ldots, M$ **do**
3:     Get two halfspaces $\mathcal{H} = \{\boldsymbol{x}|\boldsymbol{w}_m^T \boldsymbol{x} + b_m \leq 0\}$ and $\bar{\mathcal{H}} = \{\boldsymbol{x}|\boldsymbol{w}_m^T \boldsymbol{x} + b_m \geq 0\}$.
4:     **if** $\mathcal{H} \cap \mathcal{B} \neq \emptyset$ and $\hat{\mathcal{H}} \cap \mathcal{B} \neq \emptyset$ **then**
5:         Add $m$ to $\mathcal{T}$.
6: Return $\mathcal{T}$.

---

The lines marked in red are the hyperplanes cutting through the bounded region and are identified by the pre-screening algorithm. The evolution of the BFS queue is shown in Figure 2.(c). The gray arrows show the traversing order. The colored arrows at the bottom indicate the one-adjacent neighbors added to the queue. When polytope No.0 is popped from the queue, its one-adjacent neighbors, No.1, 2, 3, and 4, are added to the queue. Next, when polytope No.1 is popped, its one-adjacent neighbors, No.5 and 6, are added. Polytope No.0, although as a one-adjacent neighbor to No.1, is ignored since it has been visited. Similarly, when polytope No.2 is popped, only one of its one-adjacent neighbors, No. 7, is added, since all others have been visited (including those in the queue). The algorithm finished after popping Polytope No.8 as no new polytopes can be added and the queue is empty. All 8 local polytopes in the bounded region are traversed.

Because $\mathcal{B}$ is bounded by a set of linear inequalities, the correctness of BFS-based polytope traversing as stated in Theorem 4.1 can be easily extended to this bounded traversing case. It can be proved by showing that for any two non-empty polytopes overlapped with $\mathcal{B}$, we can move from one to another by repetitively finding a one-adjacent neighbor within $\mathcal{B}$. We emphasis that the correctness of BFS-based polytope traversing can be proved for any traversing region bounded by a set of linear inequalities. This realization is critical to generalize our results to the case of ReLU NNs with multiple hidden layers. Furthermore, as any closed convex set can

phase-I problem of LP. We name this technique **hyperplane pre-screening** and summarize it in algorithm 2.

Hyperplane pre-screening effectively reduces the complexity from $\mathcal{O}\left(2^M\right)$ to $\mathcal{O}\left(2^{|\mathcal{T}|}\right)$, where $|\mathcal{T}|$ is the number of hyperplanes cutting through the hypercube. The number $2^{|\mathcal{T}|}$ corresponds to the worst-case scenario. Since the BFS-based traversing only checks non-empty polytopes and their potential one-adjacent neighbors, the number of activation patterns actually checked can be less than $2^{|\mathcal{T}|}$. In general, the fewer hyperplanes go through $\mathcal{B}$ the faster polytope traversing finishes.

Figure 2.(a) shows traversing the 8 local polytopes within the bounded region. The ReLU NN is the same as in Figure 1.(b).

be represented as the intersection of a set of (possibly infinite) half-spaces, the correctness of BFS-based polytope traversing is true for any closed convex $\mathcal{B}$.

## 4.3 Hierarchical polytope traversing in the case of multiple hidden layers

The BFS-based polytope traversing algorithm can be generalized to ReLU NNs with multiple hidden layers. In section 2.2, we described how a ReLU NN with $L$ hidden layers hierarchically partition the input space into polytopes of $L$ different level. Then in section 3, we showed the adjacency of level-$l$ polytopes is conditioned on all of them belonging to the same level-$(l-1)$ polytope. Therefore, to traverse all level-$L$ polytopes, we need to traverse all level-$(L-1)$ polytopes and within each of them traversing the sub-polytopes by following the one-adjacent neighbors.

The procedure above leads us to a recursive traversing scheme. Assume a ReLU NN with L hidden layers and a closed convex traversing region $\mathcal{B}$. Starting from a sample $\boldsymbol{x} \in \mathcal{B}$, we traverse all level-1 polytopes using the BFS-based algorithm. Inside each level-1 polytope, we traverse all the contained level-2 polytopes, so on and so forth until we reach the level-L polytopes. As shown in (13), each level-$l$ polytope is constrained by $\sum_{t=1}^{l} M_t$ linear inequalities, the way to identify level-$l$ one-adjacent neighbors is largely the same as what we have described in Section 4.1. Two level-$l$ one-adjacent neighbors must have the same $\sum_{t=1}^{l-1} M_t$ linear inequalities corresponding to $\boldsymbol{c}^1 \boldsymbol{c}^2 \dots \boldsymbol{c}^{l-1}$, and have one of the last $M_l$ inequalities differ in direction, so there are $M_l$ cases to check.

We can use hyperplane pre-screening at each level of traversing. When traversing the level-$l$ polytopes within in a level-$(l-1)$ polytope $\mathcal{R}^{l-1}$, we update the bounded traversing region by taking the intersection of $\mathcal{R}^{l-1}$ and $\mathcal{B}$. We then screen the $M_l$ partitioning hyperplanes and only select those passing through this update traversing region.

The BFS-based hierarchical polytope traversing algorithm is summarized in Algorithm 3. The correctness of this algorithm can be proved based on the results in Section 4.2, which guarantees the thoroughness of traversing the level-$l$ polytopes within in any level-$(l-1)$ polytope. Then the overall thoroughness is guaranteed because each level of traversing is thorough. We state the result in the following theorem.

**Theorem 4.2** *Given a ReLU NN with L hidden layers and a closed convex traversing region $\mathcal{B}$. Algorithm 3 covers all non-empty level-L polytopes created by the neural network that overlap with $\mathcal{B}$. That is, for all $\boldsymbol{x} \in \mathcal{B}$, there exists one $\mathcal{R}_{\boldsymbol{c}}$ as defined in (13) such that $\boldsymbol{x} \in \mathcal{R}_{\boldsymbol{c}}$ and $\boldsymbol{c} \in \mathcal{S}_R$, where $\mathcal{S}_R$ is the result returned by Algorithm 3.*

Figure 2.(b) shows traversing the 6 local polytopes within the bounded region. The ReLU NN is the same as in Figure 1.(c). The evolution of the hierarchical BFS queue is shown in Figure 2.(d). The level-1 BFS queue is shown vertically while the level-2 BFS queue is shown horizontally. Starting from level-1 polytope $(0, \cdot)$, the algorithm traverses the two level-2 polytopes inside it (line 10 in Algorithm 3). It then identifies the two (level-1) one-adjacent neighbors of $(0, \cdot)$: $(1, \cdot)$ and $(2, \cdot)$. Every time a level-1 polytope is identified, the algorithm goes into it to traverse all the level-2 polytopes inside (line 36). At the end of the recursive call, all 6 local polytopes in the bounded region are traversed.

---

**Algorithm 3: BFS-Based Hierarchical Polytopes Traversing in a Bounded Region**

**Require:** A ReLU NN with $L$ hidden layers.
**Require:** A closed convex traversing region $\mathcal{B}$.
**Require:** An initial point $\boldsymbol{x} \in \mathcal{B}$.
 1: Initialize an empty set $\mathcal{S}_R$ to store the codes of all visited polytopes.
 2:
 3: **function** HIERARCHICAL_TRAVERSE($\boldsymbol{x}, l$)
 4:     Initialize an empty queue $\mathcal{Q}^l$ for BFS at level $l$.
 5:     Initialize an empty set $\mathcal{S}_{\boldsymbol{c}}^l$ to store all checked level-$l$ codes.
 6:     Calculate $\boldsymbol{x}$'s initial polytope code $\boldsymbol{c}$ recursively using (12).
 7:     **if** $l == L$ **then**
 8:         Add $\boldsymbol{c}$ to $\mathcal{S}_R$
 9:     **else**
10:         HIERARCHICAL_TRAVERSE($\boldsymbol{x}$,$l$+1)
11:     **if** $l > 1$ **then**
12:         Get the level-$(l-1)$ polytope code specified by the front segment of $\boldsymbol{c}$: $\boldsymbol{c}^{1:l-1} = \boldsymbol{c}^1 \boldsymbol{c}^2 \dots \boldsymbol{c}^{l-1}$.
13:         Use $\boldsymbol{c}^{1:l-1}$ to get the level-$(l-1)$ polytope $\mathcal{R}_{\boldsymbol{c}}^{l-1}$ as in (13).
14:     **else**
15:         $\mathcal{R}_{\boldsymbol{c}}^0 = \mathbb{R}^P$
16:     Form the new traversing region $\mathcal{B}^{l-1} = \mathcal{B} \cap \mathcal{R}_{\boldsymbol{c}}^{l-1}$.
17:     Append the code segment $\boldsymbol{c}^l$ to the end of the $\mathcal{Q}^l$.
18:     Add the code segment $\boldsymbol{c}^l$ to $\mathcal{S}_{\boldsymbol{c}}$.
19:     Get the $M_l$ hyperplanes associated with $\boldsymbol{c}^l$.
20:     Pre-Screen the hyperplanes associated with $\boldsymbol{c}^l$ using Algorithm 2 with bounded region $\mathcal{B}^{l-1}$.
21:     Collect the pre-screening results $\mathcal{T}$.
22:     **while** $\mathcal{Q}^l$ is not empty **do**
23:         Pop the first element in the front of BFS queue: $\boldsymbol{c}^l = \mathcal{Q}^l$.pop().
24:         **for** $m \in \mathcal{T}$ **do**
25:             Create a candidate polytope code $\hat{\boldsymbol{c}}^l$ by flipping one bit in $\boldsymbol{c}^l$: $\hat{c}_m^l = 1 - c_m^l$ and $\hat{c}_k^l = c_k^l \forall k \neq m$.
26:             **if** $\hat{\boldsymbol{c}}^l \notin \mathcal{S}_{\boldsymbol{c}}$ **then**
27:                 Get set $\mathcal{R}_{\hat{\boldsymbol{c}}} = \{\boldsymbol{x}|(-1)^{\hat{c}_k} \left( \langle \hat{\boldsymbol{w}}_k^l, \boldsymbol{x} \rangle + \hat{b}_k^l \right) \leq 0, \; k = 1, 2 \dots, M_l\}$
28:                 Check if $\mathcal{R}_{\hat{\boldsymbol{c}}} \cap \mathcal{B}^{l-1}$ is empty using LP.
29:                 Add $\hat{\boldsymbol{c}}^l$ to $\mathcal{S}_{\boldsymbol{c}}$.
30:                 **if** $\mathcal{R}_{\hat{\boldsymbol{c}}} \cap \mathcal{B}^{l-1} \neq \emptyset$ **then**
31:                     Append $\hat{\boldsymbol{c}}^l$ to the end of the $\mathcal{Q}^l$.
32:                     **if** $l == L$ **then**
33:                         Add $\hat{\boldsymbol{c}} = \boldsymbol{c}^1 \boldsymbol{c}^2 \dots \hat{\boldsymbol{c}}^l$ to $\mathcal{S}_R$
34:                     **else**
35:                         Find a point $\hat{\boldsymbol{x}} \in \mathcal{R}_{\hat{\boldsymbol{c}}} \cap \mathcal{B}^{l-1}$
36:                         HIERARCHICAL_TRAVERSE($\hat{\boldsymbol{x}}$,$l$+1)
37:
38: HIERARCHICAL_TRAVERSE($\boldsymbol{x}$,1)
39: Return $\mathcal{S}_R$.

---

# 5 The Applications of Polytope Traversing

The biggest advantage of the polytope traversing algorithm is its ability to be adapted to solve many different problems of practical interest. Problems such as local adversarial attacks, searching for counterfactual samples, and local monotonicity verification can be solved easily when the model is linear. As we have shown in Sections 2.2, the local model within each level-$L$ polytope created by a ReLU NN is indeed linear. By aggregating the results from the linear model in each polytope, our algorithm provides a way to

analyze the behavior within the entire traversing region.

The aforementioned applications have been traditionally solved using MIP (Anderson et al. 2020; Fischetti and Jo 2017; Liu et al. 2020; Tjeng, Xiao, and Tedrake 2018; Weng et al. 2018). Our algorithms based on polytope traversing have several advantages. First, our method exploits the topological structure created by ReLU NNs and fully explains the model behavior in small neighborhoods. For the $2^M$ cases created by a ReLU NN with $M$ neurons, MIP eliminates the searching branches using branch-and-bound. Our method, on the other hand, eliminates the searching branches by checking the feasibility of the local polytopes and their adjacency. Since a small traversing region often covers a limited number of polytopes, our algorithm has short running time when solving local problems.

Second, since our algorithm explicitly identifies and visits all the polytopes, the final results contain not only the optimal solution but also the whole picture of the model behavior, providing explainability to the often-so-called black-box model.

Last but probably the most important, our algorithm is highly versatile and flexible. Within each local polytope, the model is linear, which is often the simplest type of model to work with. Any analysis that one runs on a linear model can be transplanted here and wrapped inside the polytope traversing algorithm.

## 6    Conclusion

We explored the unique topological structure that ReLU NNs create in the input space; identified the adjacency among the partitioned local polytopes; developed a traversing algorithm based on this adjacency; and proved the thoroughness of polytope traversing. Our polytope traversing algorithm could be extended to other piecewise linear networks such as those containing convolutional or maxpooling layers.

## 7    Acknowledgments

The authors would like to thank Lin Dong, Linwei Hu, Rahul Singh, and Han Wang from Wells Fargo, and Sihan Zeng from Georgia Institute of Technology for their valuable inputs and feedback on this project.

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

# 8 Appendix

## 8.1 Proof of Lemma 3.1

**Lemma 8.1** *Given a set $\mathcal{R} = \{\boldsymbol{x}|g_1(\boldsymbol{x}) \leq 0, \ldots, g_M(\boldsymbol{x}) \leq 0\} \neq \emptyset$, then $g_m(\boldsymbol{x})$ is a redundant inequality if the new set formed by flipping this inequality is empty: $\hat{\mathcal{R}} = \{\boldsymbol{x}|g_1(\boldsymbol{x}) \leq 0, \ldots, g_m(\boldsymbol{x}) \geq 0, \ldots, g_M(\boldsymbol{x}) \leq 0\} = \emptyset$.*

**Proof:** Let $\tilde{\mathcal{R}}$ be the set formed by removing inequality $g_m(\boldsymbol{x}) \leq 0$: $\tilde{\mathcal{R}} = \{\boldsymbol{x}|g_1(\boldsymbol{x}) \leq 0, \ldots, g_{m-1}(\boldsymbol{x}) \leq 0, g_{m+1}(\boldsymbol{x}) \leq 0, \ldots, g_M(\boldsymbol{x}) \leq 0\}$. Then $\tilde{\mathcal{R}} = \mathcal{R} \cup \hat{\mathcal{R}}$. If $\hat{\mathcal{R}} = \emptyset$, then $\mathcal{R} = \tilde{\mathcal{R}}$ and the inequality $g_m(\boldsymbol{x}) \leq 0$ satisfies Definition 3.1. $\square$

Note the other direction of Lemma 3.1 may not hold. One example is when identical inequalities appear in the set: both inequalities in $\mathcal{R} = \{\boldsymbol{x}|g_1(\boldsymbol{x}) \leq 0, g_2(\boldsymbol{x}) \leq 0\}$ are redundant by definition if $g_1(\cdot) = g_2(\cdot)$. However, the procedure in Lemma3.1 will not identify them as redundant.

## 8.2 Proof of Theorem 4.1

**Theorem 8.2** *Given a ReLU NN with one hidden layer of $M$ neurons as specified in (1), Algorithm 1 covers all non-empty local polytopes created by the neural network. That is, for all $\boldsymbol{x} \in \mathbb{R}^P$, there exists one $\mathcal{R}_{\boldsymbol{c}}$ as defined in (3) such that $\boldsymbol{x} \in \mathcal{R}_{\boldsymbol{c}}$ and $\boldsymbol{c} \in \mathcal{S}_R$, where $\mathcal{S}_R$ is the result returned by Algorithm 1.*

**Proof:** Since each partitioning hyperplane divide $\mathbb{R}^P$ into two halfspaces, all $2^M$ activation patterns encoded by $\boldsymbol{c}$ covers the entire input space. We construct a graph with $2^M$ nodes, each representing a possible polytope code. Some the nodes may correspond to an empty set due to conflicting inequalities. For each pair of non-empty polytope that are one-adjacent to each other, we add an edge to their corresponding nodes. What left to prove is that any pair of non-empty polytopes are connected.

W.l.o.g. assume two nodes with code $\boldsymbol{c}$ and $\hat{\boldsymbol{c}}$ that differ only in the first $K$ bits. Also assume the polytopes $\mathcal{R}_{\boldsymbol{c}}$ and $\mathcal{R}_{\hat{\boldsymbol{c}}}$ are both non-empty. We will show that there must exist a non-empty polytope $\mathcal{R}_{\tilde{\boldsymbol{c}}}$ that is one-adjacent to $\mathcal{R}_{\boldsymbol{c}}$ with code $\tilde{\boldsymbol{c}}$ different from $\hat{\boldsymbol{c}}$ in one of the first $K$ bits. As a result, $\tilde{\boldsymbol{c}}$ is now one bit closer to $\hat{\boldsymbol{c}}$.

We prove the claim above by contradiction. Assuming claim is not true, we flip any one of the first $K$ bits in $\mathcal{R}_{\boldsymbol{c}}$, and the corresponding polytope $\mathcal{R}_{\tilde{\boldsymbol{c}}^k}$ must be empty. By Definition 3.1, the inequality $(-1)^{c_m}\left(\boldsymbol{w}_m^T\boldsymbol{x} + b_m\right) \leq 0, m = 1, 2, \ldots, K$ must all be redundant, which means they can be removed from the set of constraints (Telgen 1982, 1983):

$$
\begin{aligned}
\mathcal{R}_{\boldsymbol{c}} &= \{\boldsymbol{x}|(-1)^{c_m}\left(\boldsymbol{w}_m^T\boldsymbol{x} + b_m\right) \leq 0,\ m = 1, 2\ldots, M\} \\
&= \{\boldsymbol{x}|(-1)^{c_m}\left(\boldsymbol{w}_m^T\boldsymbol{x} + b_m\right) \leq 0,\ m = K+1, \ldots, M\} \\
&\supseteq \{\boldsymbol{x}|(-1)^{c_m}\left(\boldsymbol{w}_m^T\boldsymbol{x} + b_m\right) \leq 0,\ m = 1, 2, \ldots, M\} \cup \\
&\quad \{\boldsymbol{x}|(-1)^{c_m}\left(\boldsymbol{w}_m^T\boldsymbol{x} + b_m\right) \geq 0,\ m = 1, \ldots, K, \\
&\quad\quad (-1)^{c_m}\left(\boldsymbol{w}_m^T\boldsymbol{x} + b_m\right) \leq 0,\ m = K+1, \ldots, M\} \\
&= \mathcal{R}_{\boldsymbol{c}} \cup \mathcal{R}_{\hat{\boldsymbol{c}}} .
\end{aligned}
\tag{15}
$$

The derived relationship in (15) plus the assumption that all $\mathcal{R}_{\tilde{\boldsymbol{c}}^k}$ must be empty lead to the conclusion that $\mathcal{R}_{\hat{\boldsymbol{c}}} = \emptyset$, which contradict with the non-empty assumption.

Therefore, for any two non-empty polytopes $\mathcal{R}_{\boldsymbol{c}}$ and $\mathcal{R}_{\hat{\boldsymbol{c}}}$, we can create a path from $\mathcal{R}_{\boldsymbol{c}}$ to $\mathcal{R}_{\hat{\boldsymbol{c}}}$ by iteratively finding an intermediate polytope whose code is one bit closer to $\hat{\boldsymbol{c}}$. Since the polytope graph covers all input space and all non-empty polytopes are connected, BFS guarantees the thoroughness of traversing. $\square$