# OpenReview forum: "Traversing the Local Polytopes of ReLU Neural Networks"
_AAAI.org/2022/Workshop/AdvML — AAAI-22 AdvML Workshop LongPaper_

### Official Review · Reviewer_9CmL · 2021-11-24
**A novel work for proposing polytope traversing algorithm**

**Rating:** 7
**Confidence:** 3

**Review:**

This paper points out that ReLU NNs will divide the input domain into many local polytopes. Based on this observation, this work develop a polytope traversing algorithm via BFS and apply in many aspects like local adversarial attacks. The idea is novel and exactly makes sense.

However, I have some concerns about how this work can be applied to larger and more real tasks like image classification. First, the dimension of images is large, which may significantly increase the computational complexity of this algorithm. Second, since the sizes of different local polytopes may vary a lot, directly search polytopes via BFS may not find the optimal solution in some application like generating adversarial noises. Maybe the authors can attempt to test their methods in some image dataset like MNIST.

---

### Official Review · Reviewer_m27H · 2021-11-30
**Novel method for network verification**

**Rating:** 7
**Confidence:** 3

**Review:**

The author proposed a polytope traversing algorithm for network verification within a certain region. For Relu networks, the function is piecewise linear and decision regions are partitioned by many polytopes. By traversing these polytopes using a specific algorithm, we can verify a sample within a given region. I also have some concerns. What's the complexity of the method? Is it possible to scale it to larger datasets like ImageNet? I think the scalability is the major drawback of these deterministic verification approaches compared with probabilistic approaches like Randomized Smoothing. If this method can be further extended to larger-scale datasets, I think it will be a breakthrough.

---

### Decision · Program_Chairs · 2021-12-01

**Decision:**

Accept (Long Paper)

**Comment:**

Both reviewers agree to accept this paper. Please address their comments in the final version.